# A Tyrosine Kinase Expression Signature Predicts the Post-Operative Clinical Outcome in Triple Negative Breast Cancers

**DOI:** 10.3390/cancers11081158

**Published:** 2019-08-13

**Authors:** Alexandre de Nonneville, Pascal Finetti, José Adelaide, Éric Lambaudie, Patrice Viens, Anthony Gonçalves, Daniel Birnbaum, Emilie Mamessier, François Bertucci

**Affiliations:** 1Department of Medical Oncology, Institut Paoli-Calmettes, Aix-Marseille Univ, CRCM, CNRS, INSERM, 13000 Marseille, France; 2Laboratory of Predictive Oncology, Centre de Recherche en Cancérologie de Marseille, Institut Paoli-Calmettes, Inserm UMR1068, CNRS UMR725, Aix-Marseille Université, 13000 Marseille, France; 3Department of Surgical Oncology, Institut Paoli-Calmettes, Aix-Marseille Univ, CNRS, INSERM, CRCM, 13000 Marseille, France

**Keywords:** triple negative breast cancer, tyrosine kinase, gene expression signature, prognosis

## Abstract

Triple negative breast cancer (TNBC) represent 15% of breast cancers. Histoclinical features and marketed prognostic gene expression signatures (GES) failed to identify good- and poor-prognosis patients. Tyrosine kinases (TK) represent potential prognostic and/or therapeutic targets for TNBC. We sought to define a prognostic TK GES in a large series of TNBC. mRNA expression and histoclinical data of 6379 early BCs were collected from 16 datasets. We searched for a TK-based GES associated with disease-free survival (DFS) and tested its robustness in an independent validation set. A total of 1226 samples were TNBC. In the learning set of samples (N = 825), we identified a 13-TK GES associated with DFS. This GES was associated with cell proliferation and immune response. In multivariate analysis, it outperformed the previously published GESs and classical prognostic factors in the validation set (N = 401), in which the patients classified as “low-risk” had a 73% 5-year DFS versus 53% for “high-risk” patients *(p* = 1.85 × 10^−3^). The generation of 100,000 random 13-gene signatures by a resampling scheme showed the non-random nature of our classifier, which was also prognostic for overall survival in multivariate analysis. We identified a robust and non-random 13-TK GES that separated TNBC into subgroups of different prognosis. Clinical and functional validations are warranted.

## 1. Introduction

Breast cancer (BC) is heterogeneous. The latest therapeutic advances have transformed the prognosis of patients with endocrine receptors (ER)-positive and ERBB2-positive BCs [1]. BCs that do not express ER nor ERBB2, known as triple negative breast cancer (TNBCs), represent 15%–20% of BCs [2] and are high-grade tumors with poor prognosis [3]. Patients with TNBC have benefited less from recognized molecular targets than patients with other subtypes, and chemotherapy remains the only systemic treatment currently approved in the adjuvant setting, underlining the urgent need to further understand the intrinsic molecular biology of this subtype. TNBCs display heterogeneity at multiple levels, with different pathological types such as ductal or medullary [4], different intrinsic molecular subtypes [5,6] such as basal mainly but also non-basal [7], and different probabilities of relapse [8,9] and of therapeutic response [10]. Regarding the prediction of post-operative clinical outcome in early stages, both standard clinicopathological features and current gene expression signatures (GES), such as Mammaprint or Recurrence Score, failed to identify TNBC patients who will relapse and those who will not respond to chemotherapy [11]. Defining the molecular bases of this heterogeneity should help better understand these tumors and identify new therapeutic targets and more reliable predictors of survival and response. During the last years, immune signatures have been reported associated with metastatic risk and response to chemotherapy in basal and/or TNBC [12,13,14,15,16], but none of them is currently applied in clinical practice. Six and then four distinct molecular subtypes clinically and biologically relevant were identified by gene expression profiling [17,18].

Tyrosine kinases (TKs) are commonly activated in cancers and constitute major targets for anti-cancer therapies, as well as prognostic and predictive markers for therapeutic response [19]. In BC, TK activation as a driving oncogenic event has been clearly described with *ERBB2* amplification, and prognostic kinase-based GESs have been reported in basal BCs [9], luminal BCs [20], and ER-negative BCs including ERBB2-positive tumors and basal tumors [21]. Nevertheless, the aberrant TK signaling, the subjacent mechanisms, and the clinical impact in TNBC remain elusive [22,23]. Although TKs appear as potential candidates for personalized medicine, the plasticity and redundancy of the kinome present key challenges for drug development. Targeting this pathway can result in the upregulation and system-wide changes in multiple TK expression and activity as an adaptive response [24] and could explain why TNBC clinical trials of single kinase inhibitors have largely failed. Nevertheless, an approach aiming to predict clinical outcome of early stage TNBC based on TK genes expression has never been applied to a large series of TNBCs. Here, we have tested the hypothesis that a TK-gene expression signature could help the discrimination between good- and poor-prognosis patients with TNBCs.

## 2. Results

### 2.1. Patient Population

We gathered 16 retrospective public whole-genome mRNA expression data sets of 6379 operated primary BC samples, and focused our analysis on the 1226 TNBC samples with available survival. As shown in Table 1, most of the patients with TNBC were 50-years-old or more (55%), and most of the tumors were high grade (81% of grade 3), ductal type (82%), over 2 cm-size (66% of pT2-pT3), and pN-negative (59%). All Lehmann’s intrinsic molecular subtypes were represented with mainly mesenchymal, immunomodulatory, and basal like-1 subtypes (25%, 21%, and 18%, respectively). The median follow-up after diagnosis was 44 months (range, 1–286). A total of 410 patients displayed a DFS event, and the 5-year DFS was 63% (95% Confidence Interval (95%CI) 60–66). A total of 215 patients died, and the 5-year OS was 73% (95%CI 70–76).

### 2.2. Identification of a Robust Prognostic Tyrosine Kinase Signature

We searched for a gene expression signature associated with DFS within the list of 86 tyrosine kinase genes (Appendix A). As shown in Figure 1, the supervised analysis, done in the learning set of 825 TNBC samples, identified 25 genes associated with DFS (Appendix A), of which 13 were retained after Akaike information criterion (AIC) stepwise regression analysis. These 13 genes are listed in Figure 2A and were included in our prognostic classifier. As expected, this 13-gene classifier displayed prognostic value in the learning set (Figure 2B): the 5-year DFS was 52% (95%CI 46–58) in the “high-risk” class (N = 406) and 74% (95%CI 69–79) in the “low-risk” class (N = 419; *p* = 3.9 × 10^−4^, log-rank test). The respective median DFS were 70 and 238 months. Importantly, this prognostic value was confirmed in the independent validation set, suggesting the robustness of our classifier (Figure 2C): the 5-year DFS was 55% (95%CI 48–64) in the “high-risk” class (N = 206) and 71% (95%CI 64–79) in the “low-risk” class (N = 195; *p* = 1.85 × 10^−3^, log-rank test), and the respective median DFS were 107 and 225 months. 

We then generated by a resampling scheme 100,000 random 13-gene signatures extracted from the list of 86 TK genes and tested their prognostic value in the validation set. The likelihood of our 13-gene signature as a random signature was very low (*p* = 2.60 × 10^−4^).

### 2.3. Correlations of Our 13-Gene Classification with Clinicopathological and Molecular Features

To better characterize our classifier, we searched for correlations with the clinicopathological and molecular variables of tumors in the whole data set (N = 1226), in which 612 were classified as “high-risk” and 614 as “low-risk” (Table 2). Compared to the “high-risk” class, the “low-risk” class was enriched in pathological grade 3 tumors (*p* = 4.40 × 10^−2^), in node-positive tumors (*p* = 8.30 × 10^−5^), in pT1 tumors (*p* = 2.44 × 10^−2^), and in the immunomodulatory Lehmann subtype (*p* = 3.40 × 10^−17^). There was no correlation with either other clinicopathological variables (patients’ age, pathological type) or the three major prognostic GESs (70-gene signature, Recurrence Score, and ROR-P signature). Of note, the lymphocyte infiltration, relatively simple measure of immune response, which was available only for the 199 TCGA samples, was not correlated as continuous variable (*p* = 0.459) or as binary variable (*p* = 1) with our risk classes By contrast, strong correlations existed with immunity-related signatures, including the three prognostic GES reported in ER-negative BC (IR signature, LCK signature, and Immune 28-kinase) with more predicted “poor-prognosis” patients according to these signatures in our “high-risk” class, and including signatures/metagenes reflecting the activation and/or enrichment of different types of immune cells/responses. For example, the activation scores of IFNα, IFNγ, and TNFα pathways [25] were higher in the “low-risk” class, as were the cytolytic activity score [26] and the Bindea’s signatures for T-cells, cytotoxic T-cells, CD8 + T-cells, T-helper cells, Th1-cells, and Tγδ cells, activated NK CD56dim cells and neutrophils (*p* < 1.00 × 10^−5^). Similarly, the “low-risk” class was associated with enrichment for immune cell types involved in antigen presentation, such as activated dendritic cells (aDC), DC, B-cells, and macrophages. Finally, we found a higher activation probability of the MYC, P53 and hypoxia pathways in the “high-risk” class.

### 2.4. Univariate and Multivariate Prognostic Analyses for DFS

We compared the prognostic value of our 13-gene classifier for DFS with that of other clinicopathological and molecular variables in the validation set. In univariate analysis (Table 3), four factors were associated with DFS (Wald test): the Lehmann’s subtypes (*p* = 3.79 × 10^−2^), the LCK signature (*p* = 3.16 × 10^−2^), the Bindea CD8 T-cells signature (*p* = 2.18 × 10^−2^), and our 13-gene signature (*p* = 2.09 × 10^−3^) with a hazard ratio (HR) for relapse of 1.72 (95%CI 1.22–2.44) for “high-risk” patients as compared with “low-risk” patients. The IR signature tended to be associated with DFS (*p* =5.2 × 10^−2^), whereas, as expected, the variables related to cell proliferation (pathological grade, 70-gene signature, Recurrence Score, and ROR-P signature) were not associated with DFS. In multivariate analysis, two variables remained significant, including our 13-gene signature (*p* = 1.30 × 10^−2^), suggesting independent prognostic value notably when compared with other prognostic classifiers and signatures reported in ER-negative BCs. Of note, the 13-gene signature did not show any prognostic value in the non-TN cancer samples: the 5-year DFS was 78% for the “high-risk” patients versus 79% for the “low-risk” patients (*p* = 0.15, log-rank test). 

### 2.5. Univariate and Multivariate Prognostic Analyses for Overall Survival

We compared the prognostic value of our 13-gene classifier for overall survival (OS) with that of other clinicopathological and molecular variables in the whole population. In univariate analysis (Table 4), eight factors were associated with OS (Wald test): the pathological grade (*p* = 6.80 × 10^−2^), axillary lymph node status (*p* = 2.26 × 10^−6^), and tumor size (*p* = 3.77 × 10^−3^), the Lehmann’s subtypes (*p* = 1.55 × 10^−3^), the Immune 28-kinase signature (*p* = 4.14 × 10^−4^), the LCK signature (*p* = 1.37 × 10^−4^), the Bindea CD8 T-cells signature (*p* = 2.18 × 10^−2^), and our 13-gene signature (*p* = 4.48 × 10^−6^) with a hazard ratio (HR) for death of 1.88 (95%CI 1.43–2.46) for “high-risk” patients as compared with “low-risk” patients. The 5-year OS was 64% (95%CI 59–71) for the “high-risk” patients versus 79% (95%CI 75–83) for the “low-risk” patients (*p* = 3.10 × 10^−6^, log-rank test; Figure 3). In multivariate analysis, three variables remained significant, including our 13-gene signature (*p* = 4.18 × 10^−4^), suggesting independent prognostic value.

### 2.6. Biological Processes Associated to Our 13-Gene Classification

To better explore the molecular differences between the “low-risk” and “high-risk” TNBC samples as defined by our 13-gene classifier, we searched for the genes differentially expressed between the two classes in the largest sample set (Metabric: N = 335) using supervised analysis within the whole genome data. Using stringent criteria, we identified 480 differential genes, including 333 overexpressed in “low-risk” samples and 147 overexpressed in “high-risk” samples (Figure 4A; Appendix A). As expected, this gene list accurately classified the samples from the learning set, but more importantly accurately classified also the 891 samples from the independent validation set (Figure 4B), suggesting its robustness. Ontology analysis showed that the genes overexpressed in “low-risk” samples were particularly involved in immune response, whereas those overexpressed in “high-risk” samples were involved in the process of mRNA translation into protein and cell cycle progression (Figure 4C; Appendix A). 

### 2.7. Prognostic and/or Predictive Value of Our13-Gene Classifier

To determine the link of our classifier with metastatic risk and/or with response to chemotherapy, we analyzed, within our series of 1226 patients with operated TNBC, the 668 cases who had not received any adjuvant systemic therapy. In this set, the “low-risk” patients had a longer DFS than the “high-risk” patients with 5-year DFS of 79% (95%CI 74–84) versus 60% (95%CI 54–66) respectively (*p* = 2.02 × 10^−8^, log-rank test; data not shown). 

Then, we assessed the ability of our classifier to predict for the pathological complete response (pCR) to anthracycline-based neoadjuvant chemotherapy. The pCR was defined as absence of invasive residual cancer in breast and lymph nodes removed during post-chemotherapy surgery. Information was available in our dataset for 257 patients with TNBC, including 78 cases (30%) with pCR samples and 105 “low-risk” and 152 “high-risk”. The pCR rate was similar between the “low-risk” patients (31%) and the “high-risk” patients (30%; *p* = 0.861, Fisher’s exact test). Among these 257 patients, no correlation existed between the pCR rate and the other immunity-related signatures (*p* = 0.354 for the IR signature, *p* = 0.471 for the LCK signature, *p* = 0. 522 for the Immune 28-kinase signature, *p* = 0.7 for the MHC class I signature, and *p* = 0.608 for the MHC class II signature).

## 3. Discussion

Here, we have identified a robust prognostic 13-TK gene expression signature in early-stage TNBC. The 13-gene model divided TNBC into two classes (“high-risk” and “low-risk”) with different post-operative DFS and OS. Despite its association with prognostic clinicopathological features, such classification remained an independent prognostic feature in multivariate analysis in the validation set.

We analyzed a retrospective pooled set of 1226 pre-therapeutic samples of non-metastatic and invasive primary TNBC, all informative for DFS. Such figure allowed to use a learning set and a validation set thus avoiding the problem of overfitting and allowed to apply the multivariate analysis in the validation set only. Moreover, the whole-genome transcriptional data allowed to test the prognostic value of several published gene signatures and modules relevant to breast cancer, and to search for biological alterations associated to our 13-gene classifier. 

Although associated with poor-prognosis features (grade 3, pN-positivity), the «low-risk» class was associated with longer DFS than the “high-risk” class, with a HR for DFS event of 1.72 for “high-risk” as compared with “low-risk”. As expected for TNBC that are overall highly proliferative tumors, there was no correlation of our classification with the proliferation-associated prognostic GES currently used in ER+/ ERBB2− breast cancer. By contrast, our classification was strongly associated with prognostic immunity-related signatures reported in ER-negative BC (IR signature, LCK signature, and Immune 28-kinase), and with signatures/metagenes reflecting the activation and/or enrichment of different types of immune cells/responses, such as the activation score of IFNα, IFNγ, and TNFα pathways, the cytolytic activity score, and Bindea’s signatures for immune cells involved in cytotoxic immune antitumor response. For all these immune signatures, the “low-risk” class was associated with stronger cytotoxic immune antitumor response than the “high-risk” class. For example, the “low-risk” class was enriched in the Immunomodulatory Lehmann’s subtype whereas the “high-risk” class was enriched in the Mesenchymal Lehmann’s subtype. Importantly, despite such correlations, our 13-gene classifier retained its prognostic value in multivariate analysis, not only when confronted to the CD8 T-cells signature, but also to more global signatures of cytotoxic immune antitumor response. When considering the most recent Lehmann’s subtype classification [18], the “low-risk” class was enriched in the Basal-like-1 subtype and the “high-risk” class in the Mesenchymal subtype (*p* = 5.18 × 10^−11^; data not shown). Of note, and as previously reported for many prognostic GESs [12,13,14,15,16], our 13-gene signature did not show any prognostic value in the non-TN samples, further underlining the heterogeneity of molecular subtypes.

Our signature included five members of the EPH-receptor family (*EPHA1*, *EPHA4*, *EPHA7*, *EPHB4*, and *EPHB6*), two genes of the SRC family kinases (*SRC* and *FYN*), two immune-related genes (*ITK* and *ZAP70*), *FLT1*, *ALK*, and *ERBB4* receptor TKs (RTK) and *PTK2B.* A resampling scheme randomly generating 100,000 13-gene signatures showed that this data-derived 13-gene signature represents a non-random optimal prognostic combination. It represents a prognostic bar code signature for DFS in TNBC, and whether the 13 genes are causative or even predictive of the DFS event in a biological sense or reflect another associated phenomenon remains to be explored, and functional validation is warranted not only for better understanding of disease progression, but also for identification of potential therapeutic targets.

Five genes of the signature had expression associated with shorter DFS in univariate analysis: *ALK*, which codes for Anaplastic lymphoma kinase that promotes survival via activation of signaling pathways such as PI3-kinase/AKT [30,31,32,33]; *FLT1*, which codes for VEGFR1 that plays an important role in angiogenesis and exerts proliferative activity in invasive breast cancer [34] and whose high expression in BC correlates with high-risk of relapse [35]; *EPHA4*, *EPHA7*, and *EPHB4*, which code for three members of the largest family of RTK, the EPH-receptors. These three receptors have been described in breast cancer and associated with poor prognosis, migration promotion and tumor growth [36,37,38,39,40]. The eight other genes of the signature had expression associated with longer DFS in univariate analysis: *ITK* (IL2-inducible T-cell kinase), formerly considered as an immune cell-specific protein and responsible for down-streaming the T-cell receptor [41], and *ZAP70* (zeta-chain associated protein kinase), known for its role in T-cell development and lymphocyte activation, are likely signs of a T-cell cytotoxic immune response. In contrast to *EPHA4*, *A7*, and *B4*, *EPHA1,* and *B6* were associated with longer DFS. Different studies have implicated *EPHA1* and *-B6* in opposing responses than other EPH-receptors, including cell adhesion or repulsion, support or inhibition of cell proliferation and cell migration, and progression or suppression of multiple malignancies [42,43]. *EPHB6* was described as an epithelial–mesenchymal transition suppressor in TNBC cells and increased tumor sensitivity against drug therapy in TNBC xenograft models and cell lines [39,40]. Other data indicate that *EPHA1* may play different roles during the different stages of cancer progression. Low *EPHA1* expression strongly correlates with poor survival in colorectal cancer [44] and its downregulation correlates with invasion and metastasis [45]. Moreover, knockdown of *EPHA1* by CRISPR/CAS9 promotes adhesion and motility of HRT18 CRC cells [46]. As shown by our results, the action of EPH-receptors in malignant cells could be very contradictory. This discrepancy may be partially explained by the activity of so-called kinase-dead RTKs within the EPH family [47]. *FYN* and *SRC*, as components of the SRC family, are overexpressed and activated in a large number of human malignancies and have been linked to poor prognosis and endocrine therapy resistance in non-TNBC [48,49,50,51]. Nevertheless, activation of SRC family members and downstream signaling proteins are associated with a good prognosis in other types of cancer [52] and its impact in TNBC is not clear to date. 

As written above, whether these 13 genes are causative of the DFS event in a biological sense or reflect another associated phenomenon remains to be explored. The results of our supervised analysis comparing the whole-transcriptional profiles of “high-risk” versus “low-risk” TNBC samples might provide a few insights. The correlation of our 13-gene classifier with immune features was clearly demonstrated, because ontology analysis showed that the genes overexpressed in “low-risk” samples are involved in both canonical T-cell/B-cell receptors signaling pathways (e.g., Interferon-γ, perforins or granzymes genes) and in a RAS/RAC-trafficking modulation pathway potentially involved in the induction of lymphocyte-mediated tumor cells apoptosis [53]. The whole-transcriptional profiles of “high-risk” samples clearly reflect a highly metabolic behavior with genes involved in protein translation, transcription, synthesis and cell cycle control. More than half of the overexpressed genes coded for ribosomal proteins (RP). This dysregulation of multiple RP transcripts undoubtedly evokes ribosomal stress in TNBC. Ribosomal stress interferes with p19ARF/MDM2/TP53 tumor suppressor pathway and has been described as associated with shorter survival in breast cancer, independently of the molecular subtype [54]. Several other genes involved in cell cycle regulation are of interest, such as WEE1 or CCNC. WEE1 is a central kinase that controls G/M and S phase checkpoints via the phosphorylation of CDK1 and CDK2. Inhibitors of WEE1 delay mitosis in several types of cancer and make cancer cells more susceptible to chemotherapy by inducing mitotic catastrophes [55]. In breast cancer models, combination of ATR and WEE1 inhibitors inhibits tumor cells progression and metastasis process [56]. CCNC interacts with CDK8 as components of the MEDIATOR complex, a coactivator involved in regulated gene transcription of nearly all RNA polymerase II-dependent genes, making it a target of interest in TN subtype [57]. Moreover, selective inhibitors of CCNC/CDK8 with promising drug metabolism and pharmacokinetics profile are already in clinical trials for ER-positive ERBB2-negative BC [58]. Thus, TNBC relapse appears to be dependent of a multilayered interplay between cellular proliferation, stress response, and immune response. By including both immune and other prognostically relevant biological features, our 13-gene signature model displays a better prognostic value than previously published immune signatures alone.

## 4. Materials and Methods 

### 4.1. Tumor Samples and Kinase Genes

Clinicopathological and mRNA expression data were collected from 6379 primary BC patients included in 16 retrospective data sets, including ours (Appendix A). All samples were extracted from surgery specimen of non-pretreated patients, with histologically proven non-metastatic invasive TN carcinoma, and with available clinicopathological data, and had been previously profiled using DNA microarrays or RNA-sequencing. We analyzed a total of 86 genes coding for tyrosine kinases (Appendix A), selected within the list of 771 kinase and kinase-interacting genes, based on an update of the initial kinome description [21,59].

### 4.2. Gene Expression Data Processing

Data sets were processed as previously described [60]. Briefly, for the Agilent sets, we applied quantile normalization to available processed data. Regarding the Affymetrix sets, we used Robust Multichip Average (RMA) with the non-parametric quantile algorithm as normalization parameter [61]. Quantile normalization or RMA was done in R using Bioconductor and associated packages. Data analysis required pre-analytic processing. First, we normalized each DNA microarray-based data set separately, by using quantile normalization for the available processed Agilent data, and Robust Multichip Average (RMA) [61] with the non-parametric quantile algorithm for the raw Affymetrix data. Normalization was done in R using Bioconductor and associated packages. Then, we mapped hybridization probes across the different technological platforms. We used SOURCE and NCBI EntrezGene to retrieve and update the Agilent annotations, and NetAffx Annotation files [62] for the Affymetrix annotations. The probes were then mapped according to their EntrezGeneID. When multiple probes represented the same GeneID, we retained the one with the highest variance in a particular dataset. For the TCGA data, we used the available normalized RNA-Seq data that we log_2_-transformed. Next, the batch effects were corrected across the 16 studies using z-score normalization. Briefly, for each expression value in each study separately, all values were transformed by subtracting the mean of the gene in that dataset divided by its standard deviation, mean and standard deviation (SD) being measured on luminal A samples.

Thanks to the bimodal distribution of respective mRNA expression levels and in order to avoid biases related to trans-institutional immunohistochemistry (IHC) analyses, the estrogen receptor, progesterone receptor, and ERBB2 statutes (negative/positive) of tumors were defined on transcriptional data of *ESR1*, *PGR*, and *ERBB2* genes respectively, as previously described [17]. The molecular subtypes of tumors were then defined as ER+/ERBB2− for estrogen receptor-positive and/or progesterone receptor-positive and ERBB2-negative tumors, ERBB2+ for ERBB2-positive tumors, and TN for ER-negative, PR-negative and ERBB2-negative tumors. Within the 6379 samples, we identified 1226 TNBC samples that were informative regarding the survival.

### 4.3. Gene Expression Data Analysis

The search for a prognostic tyrosine kinase signature was done within these 1226 TNBC samples and comprised several steps. First, the series was divided in two sets randomly selected: the learning set including ~2/3 of samples (N = 825) and the validation set including ~1/3 of samples (N = 401). Because of the predominance of DNA microarray-derived data (N = 1027) versus RNA-Seq-derived data (N = 199), we verified that both technologies were well balanced between the learning set (68% versus 66% respectively) and the validation set (32% versus 34% respectively). Second, each of the 86 genes was tested in the learning set for association of its expression level with disease-free survival (DFS) by using Cox regression analysis (Wald test; *p* ≤ 0.05), allowing identification of 25 genes. Third, within these 25 genes we searched for the best gene combination associated with DFS by using Akaike information criterion (AIC) stepwise regression analysis, allowing identification of a 13-gene combination. A classifier was then built form the 13-gene list and allowed defining two classes of samples defined as “high-risk” and “low-risk” Finally, the classifier was applied to the samples of the validation set in order to estimate its robustness.

Since a few studies have suggested that prognostic gene signatures might be random noise signatures [63,64], we evaluated whether our prognostic 13-gene model was not inferior to random signatures. A resampling scheme was used to generate 100,000 random 13-gene models within the 86 TK genes. Each random signature was then applied to the validation set to determine its significance level in prognostic term for DFS. We then measured the proportion of random signatures with p-value inferior to the p-value observed with our 13-gene model. 

We applied to each data set separately several multigene signatures. First, the Lehmann’s classifier [17], which defines six TNBC subtypes, including 2 basal-like (BL1 and BL2), an immunomodulatory (IM), a mesenchymal (M), a mesenchymal stem-like (MSL), and a luminal androgen receptor (LAR) subtype. We also applied the three major prognostic multigene classifiers of breast cancer: 70-gene Mammaprint signature [27], Recurrence Score [28], and Risk of Relapse score based on PAM50 subtype and proliferation ROR-P [6]. Other signatures included three immune gene signatures reported as prognostic in specific molecular subtypes of breast cancer: the Immune Response (IR) signature [12] and the LCK signature [13] in ER-negative breast cancers, and the Immune 28-kinase signature [9] in basal breast cancers. We also applied the metagenes associated with immune cell populations such as T-cells, CD8+ T-cells and B-cells defined by Palmer et al. [65], the transcriptional signatures of 24 different innate and adaptative immune cell subpopulations defined by Bindea et al. [29], the cytolytic activity score [26], the activation score of IFNα, IFNγ, and TNFα immune-related and TP53, MYC and hypoxia biological pathways [25]. 

Finally, to explore more-in-depth the biological pathways linked to our 13-gene classifier, we applied a supervised analysis by using the largest data set (Metabric: 335 samples) as learning set, and the other data sets as independent validation sets (891 samples). In the learning set, we compared the whole-genome expression profiles between the tumors classified as “high-risk” (N = 86) according to our 13-gene model and the tumors classified as “low-risk” (N = 249) using a moderated t-test with empirical Bayes statistic [66] included in the limma R packages. False discovery rate (FDR) [67] was applied to correct the multiple-testing hypothesis and significant genes were defined by the following thresholds: *p* < 1.0 × 10^−2^, *q* < 1.0 × 10^−2^, and fold change FC > |1.25×|. Ontology analysis of the resulting gene list was based on the GO biological processes of the Database for Annotation, Visualization, and Integrated Discovery (DAVID; david.abcc.ncifcrf.gov/). The robustness of the resulting 480-gene list was verified in the validation set (526 tumors classified as “high-risk” and 365 as “low-risk”) by computing for each tumor a score defined as the difference between the correlation coefficient of its 480-gene expression profile with the median 480-gene expression profile of “high-risk” samples and the correlation coefficient of its 480-gene expression profile with the median 480-gene expression profile of “low-risk” samples. This score was then compared between the “high-risk” and “low-risk” samples. 

### 4.4. Statistical Analysis

Correlations between sample groups and clinicopathological factors were calculated with the Fisher’s exact test and the *t*-test when appropriate. DFS was calculated from the date of diagnosis until date of first relapse or death using the Kaplan–Meier method, and follow-up was measured to the date of last news for event-free patients. OS was calculated from the date of diagnosis until date of death using the Kaplan–Meier method. Survival curves were compared with the log-rank test. Univariate and multivariate prognostic analyses used the Cox regression method. Univariate analyses tested classical clinicopathological factors: age (≤50 years vs. >50), pathological tumor size (pT1 vs. pT2 vs. pT3), lymph node status (pN positive vs. negative), Scarff-Bloom-Richardon (SBR) grade (1 vs. 2 vs. 3), and type (ductal vs. lobular vs. other). Analyses included also molecular classifications based on the Lehmann’s subtypes, and six prognostic GES (good vs. poor-prognosis subgroups), and the Bindea CD8 T-cells GES. Multivariate analyses tested all variables with a p-value inferior to 0.10 in univariate analysis. All statistical tests were two-sided at the 5% level of significance. Analyses were done using the survival package (version 2.30), in the R software (version 2.9.1). Our analysis adhered to the reporting recommendations for tumor marker prognostic studies (REMARK) [68].

## 5. Conclusions

In conclusion, we have identified a robust prognostic 13-TK gene signature for early TNBC that outperforms the prognostic performances of individual clinicopathological prognostic factors and published gene expression signatures in term of DFS. Tumors with a coordinated cytotoxic immune anti-tumor response display longer DFS than those without, further reinforcing the fact that immune reaction is an important component of TNBC. The strength of our results lies in five main aspects: (i) the large size of the series, which represents to our knowledge the largest prognostic gene expression study reported so far in TNBC; (ii) the persistence of its prognostic value in multivariate analysis including classical prognostic signatures; (iii) its non-random nature; (iv) the biological relevance of the signature, which suggests the potential therapeutic interest of stimulating a pro-Th-1 response; and (v) the small number of genes in the signature, which should facilitate its clinical application by using other tests applicable to formaldehyde-fixed paraffin-embedded samples such as qRT-PCR. The main limitations are the retrospective nature of the study and associated biases, the absence of functional validation of genes included in the signature, and the absence of analysis at the protein level. Thus, functional validation is warranted using cell and animal models, as well as clinical validation at the protein level in large retrospective, then prospective studies.

The potential perspectives are therapeutic. Indirectly, our 13-gene classifier could improve prognostication of TNBC. The identification of poor or good-prognosis cases within operated TNBC should help tailor the systemic treatment. Analysis of DFS within patients treated without adjuvant chemotherapy and analysis of pCR to neoadjuvant chemotherapy suggest that our 13-gene classifier is associated with relapse risk, whereas its association with response to chemotherapy deserves to be tested in larger series. Since most of TNBCs are high grade (81% in our present series), it seems difficult to avoid adjuvant chemotherapy. However, the strong DFS difference between the two prognostic classes suggests that the “high-risk” patients should need a more aggressive and different treatment than “low-risk” patients. Of note, with a 71% 5-year DFS, the “low-risk” patients still have a relatively poor-prognosis and might benefit from additional treatment such as immune therapy. Thus, rather than identifying patients as candidates to de-escalation, our classifier may help to stratify patients for future clinical trials and to better develop specific additional therapies in distinct molecular subgroups of TNBC. More directly, since the anti-tumor immune response, the MEDIATOR complex and cell cycle checkpoints seem to play pivotal roles regarding the clinical outcome, the manipulation of genes and/or pathways [69,70] interfering with their functions should provide new therapeutic weapons for treating the “high-risk” patients. Of course, all these hypotheses should be tested in prospective clinical trials, before any clinical application.

## Figures and Tables

**Figure 1 cancers-11-01158-f001:**
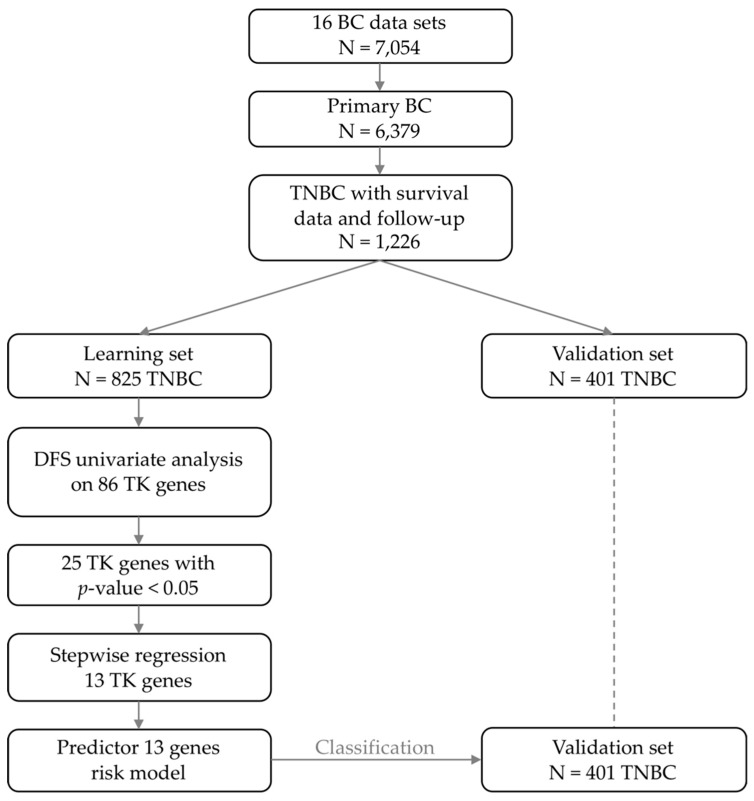
Flow diagram showing the different steps of analysis. Abbreviations: BC, breast cancer; TNBC, triple negative breast cancer; TK, tyrosine kinase; DFS, disease-free survival.

**Figure 2 cancers-11-01158-f002:**
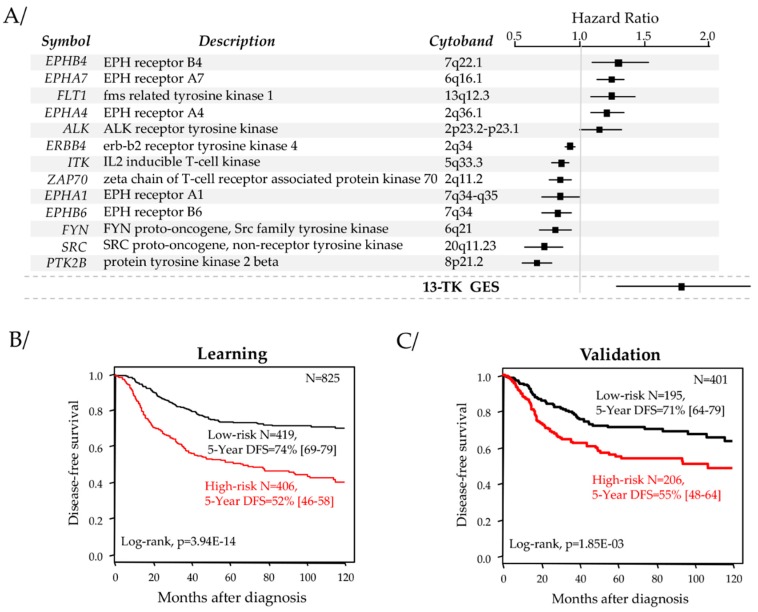
13-Gene signature and its prognostic value for DFS. (**A**) List of the 13 genes retained after Akaike information criterion (AIC) stepwise regression analysis with Forest plots of hazard ratio (HR) for DFS for each gene and the 13-TK gene expression signatures (GES). (**B**,**C**) Kaplan–Meier DFS curves of patients according to the 13-gene classifier in the learning set (**B**) and the validation set (**C**).

**Figure 3 cancers-11-01158-f003:**
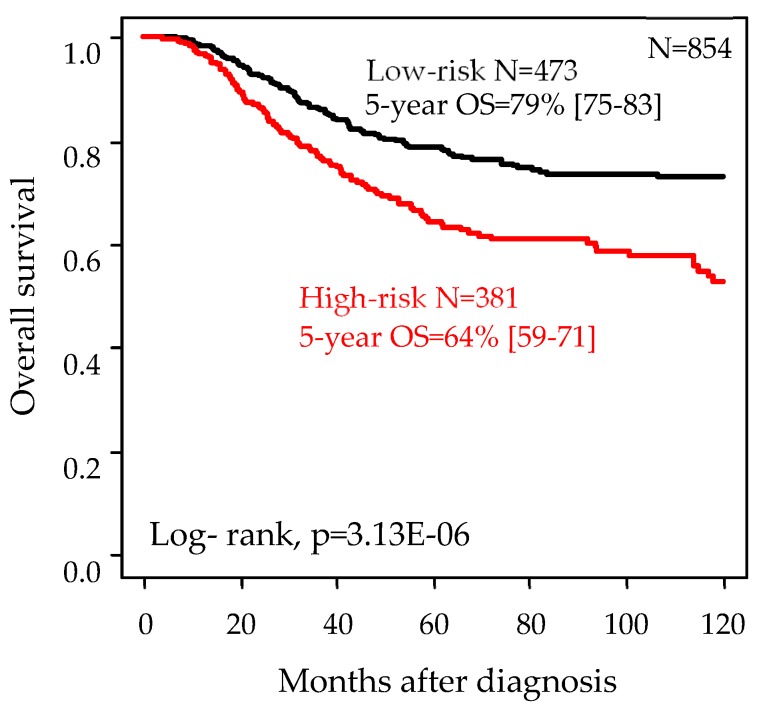
Kaplan–Meier overall survival (OS) curves according to the 13-gene classifier.

**Figure 4 cancers-11-01158-f004:**
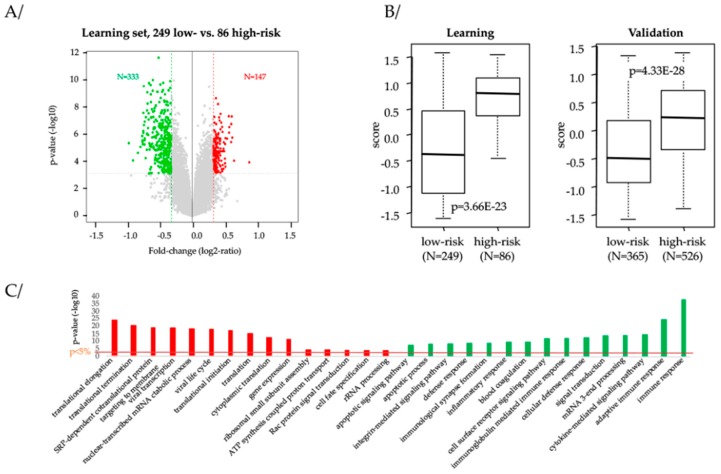
Supervised analysis of expression profiles between the “high-risk” and “low-risk” TNBC according to our 13-gene classifier. (**A**) Volcano plot showing the 480 genes differentially expressed in the learning set (Metabric), including 333 overexpressed in “low-risk” samples and 147 overexpressed in “high-risk” samples. (**B**) The metagene-based prediction score is significantly higher (Student *t*-test) in the “high-risk” samples than in the “low-risk” samples in the learning set as expected (left), but also in the independent validation set (right). (**C**) Gene ontology (GO) biological processes of the DAVID database associated with the 480-gene list. The barplot indicates the −log(*p*-value) (y-axis) of the top 15 biological pathways (x-axis) that are enriched for genes overexpressed in the “high-risk” samples versus the “low-risk” samples (in red) and that are enriched for genes overexpressed in the “low-risk” samples versus the “high-risk” samples (in green). The p-value threshold is indicated by the orange horizontal line.

**Table 1 cancers-11-01158-t001:** Description of the triple negative breast cancer (TNBC) samples.

Characteristics	N (%)
Patients’ age	
≤50 years	484 (45%)
>50 years	584 (55%)
Pathological grade	
1	23 (3%)
2	144 (17%)
3	690 (81%)
Pathological axillary lymph nodes status (pN)	
Negative	504 (59%)
Positive	347 (41%)
Pathological tumor size (pT)	
pT1	277 (34%)
pT2	459 (57%)
pT3	67 (8%)
Pathological type	
Ductal	555 (82%)
Lobular	25 (4%)
Other	98 (14%)
Lehmann’s TNBC subtypes	
Basal-like 1	216 (18%)
Basal-like 2	90 (7%)
Immunomodulatory	259 (21%)
Luminal Androgen Receptor	189 (15%)
Mesenchymal	301 (25%)
Mesenchymal stem-like	171 (14%)
Median follow-up, months (range)	44 (1–286)
DFS events	410 (33%)
5-year DFS	63% (95%CI 60–66)
OS events	215 (25%)
5-year OS	73% (95%CI 70–76)

**Table 2 cancers-11-01158-t002:** Correlations of our 13-gene classification with clinicopathological and molecular features

Characteristics		N	13-Genes Classification		*p*-Value
			Low-Risk	High-Risk	
Patients’ age					0.149
	≤50 years	484	242 (43%)	242 (48%)	
	>50 years	584	319 (57%)	265 (52%)	
Pathological grade					4.4 × 10^−2^
	1	23	14 (3%)	9 (2%)	
	2	144	69 (14%)	75 (20%)	
	3	690	408 (83%)	282 (77%)	
Pathological axillary lymph nodes status (pN)					8.3 × 10^−5^
	Negative	504	251 (53%)	253 (67%)	
	Positive	347	221 (47%)	126 (33%)	
Pathological tumor size (pT)					2.4 × 10^−2^
	pT1	277	173 (39%)	104 (29%)	
	pT2	459	242 (54%)	217 (61%)	
	pT3	67	34 (8%)	33 (9%)	
Pathological type					0.541
	Ductal	555	317 (82%)	238 (82%)	
	Lobular	25	12 (3%)	13 (4%)	
	Other	98	59 (15%)	39 (13%)	
Lehmann’s TNBC subtype [17]					3.4 × 10^−17^
	Basal−like 1	216	110 (18%)	106 (17%)	
	Basal−like 2	90	36 (6%)	54 (9%)	
	Immunomodulatory	259	188 (31%)	71 (12%)	
	Luminal Androgen Receptor	189	92 (15%)	97 (16%)	
	Mesenchymal	301	103 (17%)	198 (32%)	
	Mesenchymal stem-like	171	85 (14%)	86 (14%)	
70-gene signature [27]					0.118
	Low-risk	15	4 (1%)	11 (2%)	
	High-risk	1211	610 (99%)	601 (98%)	
Recurrence Score [28]					0.397
	Low-risk	5	1 (0%)	4 (1%)	
	High-risk	1176	591 (96%)	585 (96%)	
	Intermediate-risk	45	22 (4%)	23 (4%)	
ROR-P signature [6]					0.61
	Low-risk	77	36 (6%)	41 (7%)	
	High-risk	1005	510 (83%)	495 (81%)	
	Intermediate-risk	144	68 (11%)	76 (12%)	
Immune 28-kinase [9]					1.4 × 10^−32^
	Low-risk	298	239 (39%)	59 (10%)	
	High-risk	928	375 (61%)	553 (90%)	
IR signature [12]					6.5 × 10^−5^
	Low-risk	602	335 (56%)	267 (44%)	
	High-risk	609	268 (44%)	341 (56%)	
LCK signature [13]					2.9 × 10^−29^
	Low-risk	710	258 (42%)	452 (74%)	
	High-risk	516	356 (58%)	160 (26%)	
Gatza’s molecular pathways activation score [25]					
	IFN alpha	1226	0.68 (0.01−0.99)	0.51 (0−0.99)	3.4 × 10^−17^
	IFN gamma	1226	0.72 (0–1)	0.56 (0–0.99)	2.1 × 10^−16^
	TGF beta	1226	0.45 (0.01–1)	0.51 (0.01–1)	7.9 × 10^−5^
Lymphocyte infiltration (%)		199	8.93 (0–100)	6.99 (0–100)	0.459
Lymphocyte infiltration					1.00
	≤10%	167	58 (84%)	109 (84%)	
	>10%	32	11 (16%)	21 (16%)	
Bindea’s signatures [29]					
	B cells	1226	0.49 (−0.62–3.4)	0.14 (−0.89–1.74)	1.3 × 10^−26^
	T cells	1226	0.6 (−2.26–4.21)	0.01 (−1.6–2.09)	2.9 × 10^−37^
	T helper cells	1226	0.11 (−1.71–1.49)	0.02 (−1.11–1.61)	3.4 × 10^−5^
	Tcm	1226	0.07 (−1.28–2.19)	−0.02 (−1.09–0.82)	1.8 × 10^−6^
	Tem	1226	0.09 (−0.82–1.17)	0 (−1.07–0.88)	1.8 × 10^−7^
	Th1 cells	1226	0.27 (−0.69–1.46)	0.1 (−0.98–0.93)	4.5 × 10^−20^
	Th2 cells	1226	0.09 (−0.86–0.85)	0.12 (−0.76–1.18)	0.0734
	TFH	1226	0.03 (−1.16–1.18)	−0.06 (−0.77–0.81)	7.7 × 10^−10^
	Th17 cells	1226	−0.08 (−1.98–3.17)	−0.16 (−1.87–4.09)	0.0293
	TReg	1226	0.15 (−3.69–6.7)	0.08 (−2.35–4.42)	0.246
	CD8 T cells	1226	0.02 (−0.88–1.65)	−0.1 (−0.93–0.57)	7.5 × 10^−13^
	Tgd	1226	0.24 (−1.38–5.5)	0.02 (−2.09–2.63)	9.2 × 10^−10^
	Cytotoxic cells	1226	0.38 (−1.21–2.5)	−0.07 (−1.61–1.71)	1.9 × 10^−35^
	NK cells	1226	−0.05 (−0.96–0.89)	−0.04 (−1–0.88)	0.607
	NK CD56dim cells	1226	0.3 (−0.97–3.2)	0.08 (−1.16–2.06)	2.0 × 10^−16^
	NK CD56bright cells	1226	−0.17 (−1.49–2.9)	−0.25 (−1.69–1.25)	2.6 × 10^−4^
	DC	1226	0.31 (−1.11–3.29)	0.1 (−1.92–2.32)	3.5 × 10^−9^
	iDC	1226	0.06 (−0.89–1.61)	−0.06 (−1.26–1.58)	7.3 × 10^−9^
	aDC	1226	0.65 (−1.11–3.07)	0.24 (−1.74–4.72)	1.6 × 10^−22^
	pDC	1226	0.33 (−3.27–6.34)	−0.04 (−2.91–4.26)	2.6 × 10^−9^
	Eosinophils	1226	−0.11 (−0.84–0.73)	-0.15 (−1.01–0.55)	0.00498
	Macrophages	1226	0.24 (−1.27–2.79)	0.12 (-1.17–2.57)	1.4 × 10^−5^
	Mast cells	1226	−0.17 (-0.93–1.2)	-0.17 (-1.32–1.29)	0.96
	Neutrophils	1226	0.16 (−1.19–2.77)	0.06 (−1.22–2.19)	6.1 × 10^−5^
	SW480 cancer cells	1226	0.13 (−1.32–1.42)	0.16 (−0.77–1.59)	0.085
	Normal mucosa	1226	−0.2 (−1.4–0.96)	−0.11 (−1.59–1.31)	2.6 × 10^−4^
	Blood vessels	1226	−0.31 (−4.44–4.93)	−0.23 (−3.39−3.66)	0.169
	Lymph vessels	1226	−0.06 (−2.06–2.75)	−0.13 (−2.46−2.55)	0.0924
Cytolytic activity score [26]		1226	0.41 (−2.42–4.66)	0.05 (−1.95–3.75)	7.87 × 10^−10^
Median follow-up (months)		1226	53 (1–286)	31 (1–229)	9.8 × 10^−17^
DFS events (months)		1226	157 (74%)	253 (41%)	5.6 × 10^−9^
5-year DFS		1226	73% (69–77)	53% (48–58)	1.3 × 10^−15^
OS events (months)		1226	101 (21%)	114 (30%)	4.3 × 10^−3^
5-year OS		1226	79% (75–83)	64% (59–71)	3.0 × 10^−6^

**Table 3 cancers-11-01158-t003:** Uni- and multivariate prognostic analyses for DFS in the validation set.

Variable	Test	Univariate			Multivariate		
		N	HR (95% CI)	*p*-Value	N	HR (95%CI)	*p*-Value
Patients’ age	>50 vs. ≤50 years	351	0.88 (0.60−1.31)	0.540			
Pathological grade	2 vs. 1	275	18841300 (0–Inf)	0.445			
	3 vs. 1		27280092 (0–Inf)				
Pathological axillary lymph node status (pN)	Positive vs. negative	273	1.35 (0.87–2.09)	0.177			
Pathological tumor size (pT)	pT2 vs. pT1	254	1.35 (0.81–2.25)	0.505			
	pT3 vs. pT1		1.32 (0.50–3.50)				
Pathological type	Lobular vs. ductal	219	2.12 (0.76–5.92)	0.337			
	Other vs. ductal		1.19 (0.60–2.35)				
Lehmann’s TNBC subtype [17]	Basal-like 2 vs. Basal-like 1	401	2.44 (1.27–4.69)	3.79 × 10^−2^	394	2.62 (1.36–5.08)	4.22 × 10^−3^
	Immunomodulatory vs. Basal-like 1		0.98 (0.57–1.67)		394	1.48 (0.82–2.67)	0.198
	Luminal AR vs. Basal-like 1		1.15 (0.65–2.04)		394	1.30 (0.72–2.33)	0.385
	Mesenchymal vs. Basal-like 1		0.88 (0.52–1.52)		394	0.75 (0.43–1.29)	0.292
	Mesenchymal stem-like vs. Basal-like 1		0.85 (0.47–1.55)		394	1.16 (0.61–2.19)	0.650
70-gene signature [27]	High- vs. low-risk	401	2.88 (0.40–20.6)	0.292			
Recurrence Score [28]	High- vs. low-risk	401	1233058 (0–Inf)	0.635			
	Intermediate- vs. low-risk		624862 (0–Inf)				
PAM50 and ROR-P [6]	High- vs. low-risk	401	6.88 (0.96–49.3)	0.152			
	Median vs. low-risk		6.30 (0.84–47.3)				
28-gene Immune Kinase [9]	High- vs. low-risk	401	1.22 (0.83–1.79)	0.313			
Immune response [12]	High- vs. low-risk	394	1.41 (1.00–1.98)	0.052	394	1.33 (0.93–1.90)	0.125
Lymphocyte-specific kinase [13]	High- vs. low-risk	401	0.68 (0.48–0.97)	0.032	394	0.83 (0.52–1.33)	0.445
Bindea’s CD8 T-cells signature		401	0.47 (0.25–0.90)	2.18 × 10^−2^	394	0.51 (0.24–1.08)	0.080
13-tyrosine kinase genes	High- vs. low-risk	401	1.72 (1.22–2.44)	2.09 × 10^−3^	394	1.61 (1.11−2.36)	1.30 × 10^−2^

**Table 4 cancers-11-01158-t004:** Uni- and multivariate prognostic analyses for OS.

Variable	Test	Univariate			Multivariate		
		N	HR (95%CI)	*p*-Value	N	HR (95% CI)	*p*-Value
Patients’ age	>50 vs. ≤50 years	830	0.84 (0.64–1.10)	0.209			
Pathological grade	2 vs. 1	641	2.36 (0.56–9.93)	0.068	532	2.14 (0.5–9.24)	0.308
	3 vs. 1		3.33 (0.83–13.5)		532	2.84 (0.67–12.1)	0.158
Pathological axillary lymph node status (pN)	Positive vs. negative	742	2.00 (1.50–2.67)	2.26 × 10^−6^	532	1.93 (1.39–2.68)	8.04 × 10^−5^
Pathological tumor size (pT)	pT2 vs. pT1	744	1.59 (1.16–2.19)	3.77 × 10^−3^	532	1.34 (0.94–1.91)	0.106
	pT3 vs. pT1		2.04 (1.22–3.39)		532	1.37 (0.75–2.51)	0.299
Pathological type	Lobular vs. ductal	677	0.91 (0.42–1.94)	0.144			
	Other vs. ductal		0.60 (0.37–1.00)				
Lehmann’s TNBC subtype [17]	Basal-like 2 vs. Basal-like 1	854	1.49 (0.83–2.68)	1.55 × 10^−3^	532	1.99 (1.02–3.9)	4.47 × 10^−2^
	Immunomodulatory vs. Basal-like 1		0.54 (0.34–0.87)		532	1.04 (0.57–1.91)	0.892
	Luminal AR vs. Basal-like 1		1.12 (0.73–1.70)		532	1.67 (1.00–2.77)	4.91 × 10^−2^
	Mesenchymal vs. Basal-like 1		1.33 (0.91–1.94)		532	1.72 (1.08–2.76)	2.33 × 10^−2^
	Mesenchymal stem-like vs. Basal-like 1		0.79 (0.47–1.31)		532	1.15 (0.57–2.30)	0.699
70-gene signature [27]	High- vs. low-risk	854	3.03 (0.42–21.6)	0.269			
Recurrence Score [28]	High- vs. low-risk	854	1236427 (0–Inf)	0.42			
	Intermediate- vs. low-risk		330274 (0–Inf)				
PAM50 and ROR-P [6]	High- vs. low-risk	854	1.90 (0.84–4.28)	0.217			
	Median vs. low-risk		1.54 (0.62–3.84)				
28-gene Immune Kinase [9]	High- vs. low-risk	854	1.83 (1.31–2.55)	4.14 × 10^−4^	532	0.96 (0.53–1.71)	0.881
Immune response [12]	High- vs. low-risk	840	1.16 (0.89–1.52)	0.28			
Lymphocyte-specific kinase [13]	High- vs. low-risk	854	0.58 (0.44–0.77)	1.37 × 10^−4^	532	0.92 (0.57–1.47)	0.713
Bindea’s CD8 T-cells signature		854	0.45 (0.27–0.75)	2.16 × 10^−3^	532	0.94 (0.45–1.97)	0.87
13-tyrosine kinase genes	High- vs. low-risk	854	1.88 (1.43–2.46)	4.48 × 10^−6^	532	1.86 (1.32–2.62)	4.18 × 10^−4^

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
