# Peer review of "A Tyrosine Kinase Expression Signature Predicts the Post-Operative Clinical Outcome in Triple Negative Breast Cancers"

_cancers, 2019, doi:10.3390/cancers11081158_

Round 1
Reviewer 1 Report
This paper describes the development of a 13-gene expression signature as a prognostic tool for triple-negative breast cancer patients. The discovery method uses a fairly straightforward statistical approach. The rationale for focusing on genes that express TKs is not entirely clear, since it is TK activation, not gene expression, that drives oncogenesis. Nevertheless, there are clear examples (e.g. HER2) where overexpression is a major cause of oncogenic activity, and the authors’ approach yields a prognostic panel derived from a >800-patient discovery set, that performs well on a >400-patient validation set. Whether, as suggested in the Conclusion, there are therapeutic implications, is entirely speculative at this stage.
Specific points
1. The authors will be aware that the six Lehmann TNBC subtypes are now re-classified into 4 subtypes (PMID: 27310713), reflecting in part that the immunomodulatory subtype largely describes gene expression from infiltrating lymphocytes. This should be included in the Introduction and Discussion. Since the 13-gene signature is strongly correlated with immunity-related signatures, to what extent is its prognostic performance simply reflecting the well-established observation that a high tumor immune infiltration, in particular CD8 TILs, is prognostic for better survival?
2. Most of the data are derived from microarrays. When allocating the 1226 TNBC samples between discovery and validation groups, how were the RNA-Seq-derived data distributed between the two groups?
3. Was the gene signature also tested against overall survival?
Author Response
Reply to the Review Report is attached as a word document

Reviewer 2 Report
In the present manuscript, Alexandre DE Noneville et al. used 1226 TNBC patients’ mRNA and histoclinical data, searched TK-based GES associated with DFS and then tested its robustness. They finally identified 13 TK GESs associated with DFS in TNBC patients and could be a better prognostic marker in TNBC patiens. This is an interesting study and a well conducted one. I appreciate the study design and the observations made during the study. However, there are some questions are still needed to be clarified.
1. How to utilize the authors’ findings to clinical application? Most of the tumors were high grade (81% of grade 3), therefore, surgery followed by adjuvant chemotherapy should be a standard of treatment for locally advanced BC. Is there any correlation among these different drug responses and 13 TK-associated genes, immunity-related signatures, MHC class I/II antigen presentation pathways and immune cells infiltration?
2. The authors did not show the direct evidences by cell or animal model to proof their findings.
3. The present manuscript lacks the immunohistochemistry staining for further verification
Minor:
The abstract should be written in one paragraph.
Author Response

(The authors gave the same response as above.)

Round 2
Reviewer 2 Report
The authors have made satisfactory improvement in response to the reviewers' comments and suggestions.